# Straw Return with Reduced Nitrogen Fertilizer Maintained Maize High Yield in Northeast China

**Yanjie Lv, Yongjun Wang \*, Lichun Wang and Ping Zhu \***

State Engineering Laboratory of Maize, Institute of Agricultural Resources and Environment,
Jilin Academy of Agriculture Sciences, Changchun 130033, Jilin, China; lvyanjie@cjaas.com (Y.L.);
wanglichun@cjaas.com (L.W.)

\* Correspondence: yjwang2008@cjaas.com (Y.W.); zhuping2008@cjaas.com (P.Z.);
  Tel.: +86-431-8706-3941 (Y.W.);  +86-431-8706-3185 (P.Z.);
  Fax: +86-431-8507-2010 (Y.W.); +86-431-8706-3165 (P.Z.)

**Abstract:** Crop residue management is a major concern in China. Returning straw to the field has been promoted to reduce chemical fertilizer input and increase crop yield. For this, the present study as a part of an existing long-term field experiment was conducted including four treatments: (1) Maize straw return with reduced nitrogen, phosphorus, and potassium fertilizers (straw + NPK; SNPK); (2) NPK fertilizers; (3) PK fertilizers; and (4) no fertilizers added (nF) in the years of 2016 and 2017. Results showed SNPK and NPK produced similar grain yield over the two years (12,485 vs. 12,754 kg ha$^{-1}$), which was approximately 3-fold of PK and nF (4763 vs. 3858 kg ha$^{-1}$). The yield difference was not significant between SNPK and NPK and between PK and nF. The effect of straw return on plant productivity is related to precipitation. In contrast to total carbon (C), nitrogen (N), and phosphorus (P), K was lost from leaf, stem, and grain. Dynamics of plant N post-flowering is critical to determine maize yield and yield components relative to C, P, and K. Dynamics of P and K in leaf were more correlated with yield than in stem, implying the significance of leaf productivity in determining grain yield. These results indicate that combination of NPK fertilizers is critical for increasing grain yield in maize. Crop straw return with reduced NPK fertilizers will help to improve yield and reduce chemical fertilizer input in the long run.

**Keywords:** straw return; combined fertilizer application; NPK; maize

## 1. Introduction

Soil degradation is a major global threat to agriculture and has been of great concern to the black soil (mollisols) region of Northeast China [1,2]. Approximately 30% of China's staple food is produced in the black soils of this region [2,3], where soil degradation has been rapidly increasing due to the rise in modern agriculture since 1950 [4,5]. Therefore, effective management strategies to reduce soil degradation in this region are critical before it poses a long-term threat to food security in China.

Subsistence agriculture has caused soil nutrient depletion [6], and the practices of modern intensive farming are accelerating this process, causing a negative nutrient balance in the soil [7]. Intensive farming that involves continuous and increased use of chemical fertilizers significantly increased crop yield [8,9]; however, it had negative effects on soil quality [10,11], such as acidifying effects on soil [9]. These changes in soil quality will alter plant growth, productivity, nutrient dynamics, and yield. Maintenance of adequate fertilizer (NPK) input as well as fertilizer balance between different nutrients in the farmland is the critical factor that determines sustainability in the long perspective [9,12]. Continuous cultivation with low or high fertilizer input can affect soil quality by changing soil chemical and physical properties [13–15]. Long-term application of N, P, or K alone may not produce the

expected crop yield and may reduce fertilizer utilization efficiency [16,17]. Overusing fertilizers may lead to excessive salt and nitrate concentrations, thus reducing soil quality [18].

To better understand the soil nutrient dynamics and cope with soil degradation, several strategies were attempted worldwide. Notably, long-term field experiments are considered an effective way to investigate soil quality changes due to fertilizer application over time and to evaluate the effects of strategies for improving soil fertility [1,13,19,20]. Studies have indicated that the judicious use of inorganic fertilizers along with manure or straw can enhance soil organic matter and microbial biomass [11,19]. Compared to chemical (NPK) fertilization, organic fertilization and crop straw return with NPK supplementation are promising strategies to improve soil quality and production capacity that will increase crop yield in the long run [11,21,22]. Crop straw return to the field post-harvest is strongly encouraged in China to reduce the air pollution resulting from straw burning [23]. Long-term field experiments have proven that combining straw return and chemical fertilizer can help to achieve grain yield similar to chemical fertilizer alone [24,25]. However, the underlying mechanism remains unclear as to how straw return replaces chemical fertilizer, especially in terms of aboveground nutrient dynamics. Based on the effects of crop straw return on sustainable management of soil quality, we hypothesized that organic N contained in the crop straw can replace the equal amount of inorganic N to match the nutrient demand and to maintain crop growth and productivity, in the comparison between the scenarios of straw + reduced NPK fertilizers and NPK fertilizers. However, long-term effects of combined straw return and chemical fertilization on maize grain yield have not yet been analyzed from the perspective of plant nutrient dynamics.

Therefore, the present study evaluated and compared the effects of combined straw return and combined chemical fertilization on plant growth, plant productivity, grain yield, and nutrient dynamics in maize. This study was performed on an existing long-term field experiment that was initiated in 1990, with NPK and PK fertilizer treatments, combination of straw return and NPK fertilizers, and no fertilizer control.

## 2. Materials and Methods

### 2.1. Experimental Site and Design

The two-year experiment was conducted as part of an existing long-term different fertilizer application field experiment on black soil (black soil is one kind of soil types according to the Chinese soil classification system; it is similar to the vertosols and Mollisols) initiated in 1990 at the Gongzhuling experimental station, Jilin Academy of Agricultural Sciences, Gongzhuling City, Northeast China (43°30′ N, 124°48′ E). Mean precipitation and temperature during the growing season were approximately 550 mm and 17.0 °C, respectively. Information on weather conditions during the maize-growing season of 2016 and 2017 is shown in Figure 1. Each field experiment plot was collected in 5 replicates by soil drill before sowing. The soil total of N, P, and K was determined using standard methods [26]. Basic soil characteristics at 0–20 cm depth are shown in Table 1. Maize hybrid Zhengdan958 (provided by Institute of Crop Science, Chinese Academy of Agricultural Science) was used in the study, which has been a favorable maize hybrid for more than 10 years in China.

**Table 1.** Basic soil characteristics of the experiment.

| Treatments | SOM (g kg$^{-1}$) | TN (g kg$^{-1}$) | TP (g kg$^{-1}$) | TK (g kg$^{-1}$) | AN (mg kg$^{-1}$) | AP (mg kg$^{-1}$) | AK (mg kg$^{-1}$) |
|---|---|---|---|---|---|---|---|
| SNPK | 33.72 | 1.59 | 0.87 | 20.58 | 120.31 | 58.42 | 184.67 |
| NPK | 28.73 | 1.50 | 0.83 | 20.13 | 127.5 | 99.92 | 166.05 |
| PK | 29.04 | 1.40 | 0.93 | 20.46 | 103.83 | 119.04 | 213.32 |
| nF | 28.81 | 1.35 | 0.53 | 20.33 | 99.10 | 5.46 | 148.16 |

SOM, soil organic matter; TN, total nitrogen; TP, total phosphorus; TK, total potassium; AN, available nitrogen; AP, available phosphorus; AK, available potassium.

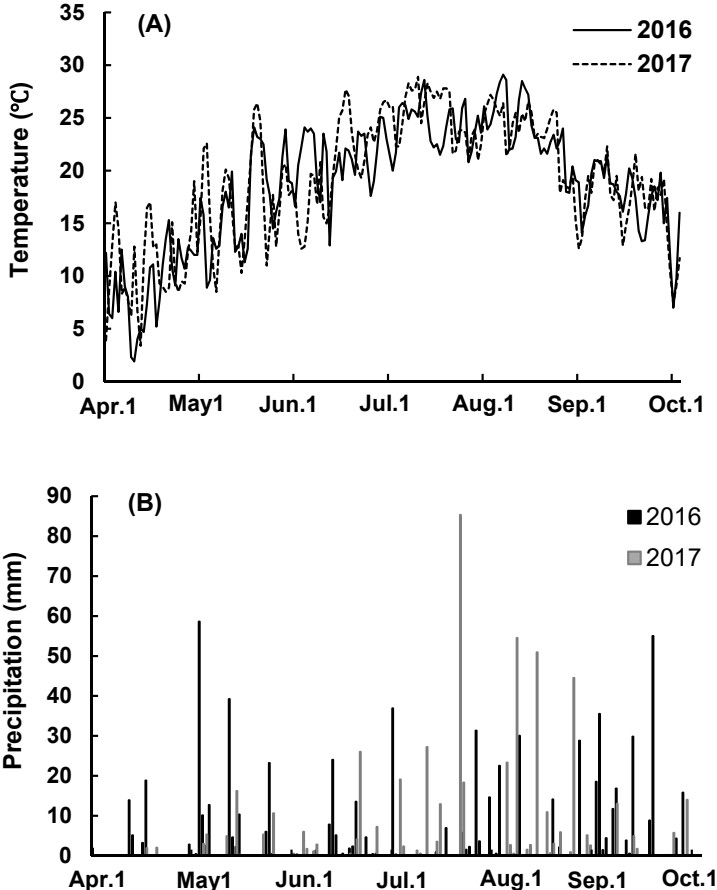

**Figure 1.** Daily temperature (**A**) and precipitation (**B**) during the maize-growing season in 2016 and 2017 at the experimental site.

We employed a randomized complete block design for the experiment under corn continuous cropping system with four fertilizer treatments as follows: (1) Maize straw, nitrogen, phosphorus, and potassium fertilizers added (Straw + NPK fertilizers); (2) nitrogen, phosphorus, and potassium fertilizers added (NPK); (3) phosphorus and potassium fertilizers added (PK); and (4) no fertilizer added (nF). Details of fertilizer treatments are given in Table 2. In SNPK and NPK treatments, the amount of N fertilizer used was 165 kg ha$^{-1}$ (dosage usually used for long-term application in this region). Straw N used in SNPK treatment was set at 53 kg ha$^{-1}$, and maize straw was broadcasted on the soil surface before sowing of maize. The amount of maize straw was calculated based on the N content of maize straw (7.57 Mg ha$^{-1}$, N content: 0.7%). N fertilizer (Urea) was applied at two-time points; one third was applied at sowing and the rest at 6-leaf stage, in both SNPK and NPK treatments. At sowing, 82.5 kg P$_2$O$_5$ ha$^{-1}$ (Superphosphate) and 82.5 kg K$_2$O ha$^{-1}$ (Potassium chloride) were applied in SNPK, NPK, and PK treatments. Each treatment had three replicates; the plot size was 6.5 m $\times$ 20 m, and row spacing was 0.65 m.

*2.2. Field Management*

After harvesting of maize, inversion tillage was conducted with a mold board plow. Seeds were not sown in cold winter until the following spring. Sowing and harvesting dates were 30 April 30 and 26 September, respectively, in 2016 and 2017. Seeds were sown using a precision seeder, which maintained single grain sowing. Plots were south–north row orientation and the final stand of 60,000 plants ha$^{-1}$. Each plot consisted of 16 rows 50 m long and with 0.65 m inter-row spacing. Plants were sampled from the four center rows. An optimal management strategy was used during the entire maize-growing season to control weeds, insects, and pathogens.

**Table 2.** Fertilizer components and amount and applied time in treatments of SNPK, NPK, PK, and nF.

| Treatments | Straw NPK [#] | | N Fertilizer | | P Fertilizer ($P_2O_5$) | | P Fertilizer ($K_2O$) | |
|---|---|---|---|---|---|---|---|---|
| | Amount (kg ha$^{-1}$) | Time | Amount (kg ha$^{-1}$) | Time | Amount (kg ha$^{-1}$) | Time | Amount (kg ha$^{-1}$) | Time |
| SNPK | Straw 7571 N 53 $P_2O_5$ 12.1 $K_2O$ 56.8 | before seeding | 112 | 1/3 at sowing and 2/3 at V6 | 82.5 | Sowing | 82.5 | Sowing |
| NPK | 0 | | 165 | 1/3 at sowing and 2/3 at V6 | 82.5 | Sowing | 82.5 | Sowing |
| PK | 0 | | 0 | | 82.5 | Sowing | 82.5 | Sowing |
| nF | 0 | | 0 | | 0 | | 0 | |

[#] SNPK (Straw NPK) represents a combination of maize straw, nitrogen, phosphorus, and potassium fertilizers, NPK represents the treatment of nitrogen, phosphorus, and potassium fertilizers, PK represents phosphorus and potassium fertilizers, and nF represents no fertilizer. N, $P_2O_5$, and $K_2O$ contents in the maize straw were 0.7%, 0.16%, and 0.75% in 2015, respectively. According to the N demand (53 kg ha$^{-1}$), the amount of maize straw that was applied after harvest of maize was 7.57 Mg ha$^{-1}$, and straw P and K were 12.1 and 56.8 kg ha$^{-1}$. The amount of straw return was calculated based on N demand due to the importance of N fertilizer, P and K of straw were considered less in the present study because of the low P content of straw and sufficient K content in soil in the experimental region. V6 represents six leaf stage (Table 1).

## 2.3. Sampling and Data Collection

### 2.3.1. Leaf Area Index

The green leaf area was determined at silking and post-silking (11, 21, 32, 42, and 50 days after silking (DAS) in 2016 (total six timings) and 19, 28, 40, and 55 DAS in 2017 (total five timings). In each plot, three plants were randomly selected, and the green leaf area was calculated Σ (leaf length × maximum leaf width) × 0.75. Bottom leaves with half yellow area were regarded as senesced.

### 2.3.2. Leaf Photosynthesis

Ear leaf net photosynthetic rate (Pn) was measured for three carefully selected plants in each plot at the silking stage at approximately 30 and 50 DAS. A portable photosynthesis system (Li6400; LI-COR, Lincoln, NE, USA) was used, and the measurements of ear leaf photosynthesis were taken between 10:00–15:00.

### 2.3.3. Plant Dry Matter

To determine the dynamics of plant dry matter, three randomly selected plants of each plot were collected from silking stage to harvest at approximately 10-day intervals. Leaves, stems, and ears were separated and dried in an oven at 60 °C until weight became stable. The plant dry matter (g plant$^{-1}$) was the sum of the dry weights of leaf, stem, and ear.

### 2.3.4. Plant Nutrient Analysis

After measuring the plant dry matter, dry leaves, stems, and grain kernels were ground separately and passed through a 1 mm sieve to analyze the nutrient concentration (%) (total C, N, P, and K). Total C and N were determined using a LECO CN analyzer (Dumas combustion). Total P and K in the nitric–perchloric tissue digests were measured using ICP-AES (Atomic Emission Spectrometer) or ICP-MS (Inductively Coupled Plasma Mass Spectrometry) [27]. Nutrient content (g plant$^{-1}$) was calculated by multiplying nutrient concentration (%) and plant dry matter.

### 2.3.5. Grain Yield and Yield Components

At maturity (kernel black layer was visible in 50% of the ears), ears in four adjacent rows of length 10 m from half of each plot were collected, counted, and weighed. The average fresh ear weight (G) was calculated, by which 20 ears were selected, with total weight approaching 20 × G. Kernels per ear were counted, threshed, and oven-dried at 80 °C to a stable dry weight. The grain yield was calculated

based on dry weight of kernels per ear, harvested area, and a number of ears in the harvested area. The grain yield was adjusted to 14% moisture content. Six samples of 500 kernels each were counted and weighed to measure the 1000-kernel weight (TKW). Harvest index (HI) was expressed as the ratio of dry grain weight over aboveground plant dry matter per unit area.

*2.4. Statistical Analysis*

Analysis of variance for grain yield and yield components was performed with PROC MIXED (9.3; SAS Institute, Cary, NC, USA). Effects of fertilizer treatment, experimental year, and their interaction was treated as the fixed factor and field replicates as the random factor. Correlations between the dynamics of total C, N, P, and K and yield were calculated in SPSS 21 (IBM, Chicago, IL, USA). Statistical significance was determined by *t*-test ($\alpha < 0.05$).

## 3. Results

*3.1. Grain Yield and Yield Components*

The effects of long-term fertilizer treatments on grain yield and yield components in maize were significant (Table 3). The experimental year showed a significant effect only on grain yield and not on kernels ear$^{-1}$ and TKW. No significant differences were detected in yield and yield components between SNPK and NPK and between PK and nF. SNPK and NPK produced similar grain yield over the two years (12485 vs. 12754 kg ha$^{-1}$), which was approximately 3-fold of PK and nF (4763 vs. 3858 kg ha $^{-1}$). Over the two years, SNPK and NPK produced 1.7-fold more kernels ear$^{-1}$ and 1.3-fold more TKW than PK and nF. The harvest indices of SNPK and NPK were also 13.0–14.4% higher compared to those of PK and nF. On average across year and treatment, the overall mean values in yield, kernel number, TKW, and HI were 8465.3 kg ha$^{-1}$, 424.6 kernel per ear, 351.3 g, and 50.5, respectively.

**Table 3.** Grain yield, kernel number per ear (kernel ear$^{-1}$), thousand kernel weight (TKW), and harvest index (HI) as the effect of long-term fertilizer treatment SNPK, NPK, PK, and nF.

| Item | Yield (kg ha$^{-1}$) | Kernel Ear$^{-1}$ | TKW (g) | HI |
|---|---|---|---|---|
| 2016 SNPK | 11464.3 (224.7) a | 570.7 (9.7) a | 384.3 (8.1) a | 0.52 (0.001) a |
| NPK | 11789.1 (224.7) a | 576.7 (9.7) a | 400.2 (8.1) a | 0.51 (0.001) a |
| PK | 4486.0 (224.7) b | 292.0 (9.7) b | 313.8 (8.1) b | 0.44 (0.001) b |
| nF | 4100.9 (224.7) b | 228.7 (9.7) b | 302.7 (8.1) b | 0.46 (0.001) b |
| *Mean* | *7960.1 (136.9)* | *417.0 (6.0)* | *350.3 (4.8)* | *0.48 (0.007)* |
| 2017 SNPK | 13505.9 (315.1) a | 581.3 (13.7) a | 431.2 (10.9) a | 0.56 (0.002) ab |
| NPK | 13719.3 (315.1) a | 596.7 (13.7) a | 381.0 (10.9) a | 0.57 (0.002) a |
| PK | 5041.3 (315.1) b | 306.0 (13.7) b | 292.4 (10.9) b | 0.51 (0.002) b |
| nF | 3615.3 (315.1) b | 244.7 (13.7) b | 304.9 (10.9) b | 0.47 (0.002) b |
| Mean | 8970.4 (136.9) | 432.2 (6.0) | 352.4 (4.8) | 0.53 (0.007) |
| Overall mean | 8465.3 (96.8) | 424.6 (4.2) | 351.3 (3.4) | 50.5 (0.5) |
| Year | *** | ns | ns | *** |
| Treatment | *** | *** | *** | *** |
| Year × Treatment | ** | ns | ** | ns |

SNPK represents a combination of maize straw, chemical nitrogen, phosphorus, and potassium, NPK represents the treatment of chemical nitrogen, phosphorus, and potassium, PK represents chemical phosphorus and potassium, and nF represents no fertilizer. Values with the same letter are not significantly different at *p* < 0.05, comparison within the same parameter only in the same year. Values in the bracket are standard error of the mean. ** and *** represent *p* < 0.01 and *p* < 0.001, respectively; ns, no significant difference at *p* = 0.05.

*3.2. Plant Dry Matter*

Plant dry matter (DM) increased from silking stage to approximate 50 DAS and then remained stable for all the treatments in both years (Figure 2). SNPK and NPK produced approximately two-fold more DM than PK and nF during the entire grain-filling period. NPK produced slightly more DM than SNPK, and the difference was significant only during the first half of the grain-filling period. PK produced more DM than nF, and the difference became larger with time during the grain-filling period.

At harvest, SNPK, NPK, PK, and nF produced, on average, 350.4, 355.5, 195.1, and 160.4 g plant$^{-1}$ DM over two years, respectively. On average, the highest DM produced in 2016 was more than in 2017 (280 vs. 270 g plant$^{-1}$).

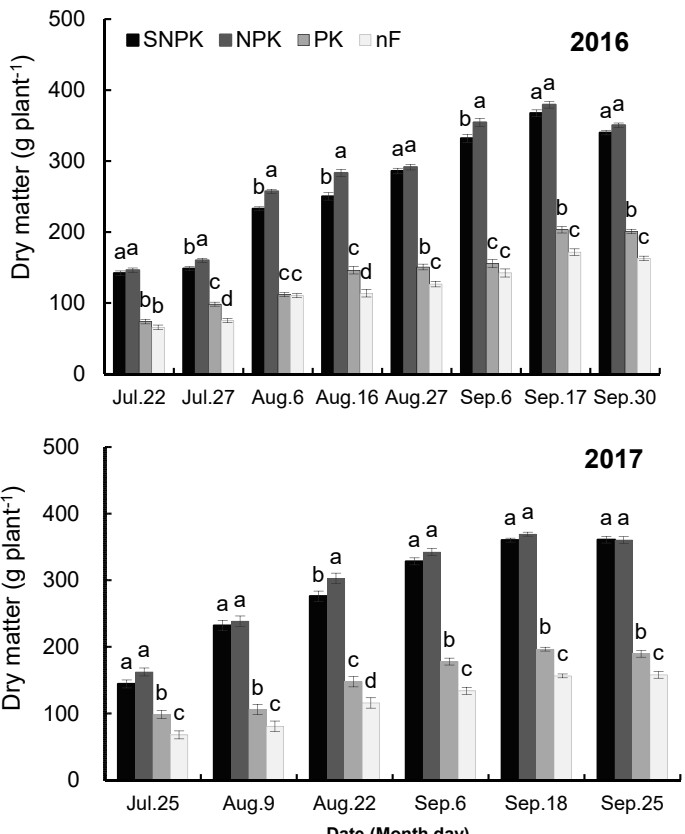

**Figure 2.** Dry matter dynamics post-flowering of single maize plant as effect of long-term fertilizer treatment SNPK, NPK, PK, and nF. SNPK represents combination of maize straw, chemical nitrogen, phosphorus, and potassium, NPK represents treatment of chemical nitrogen, phosphorus, and potassium, PK represents chemical phosphorus and potassium, and nF represents no fertilizer. Values with the same letter are not significantly different at *p* < 0.05, comparison within the same sampling date. Bars: Standard error of the mean.

### 3.3. Leaf Area Index (LAI)

Fertilizer treatment had a significant effect on the dynamics of LAI after silking stage (Figure 3). SNPK recorded a maximum LAI close to NPK (4.5 vs. 5.0 in 2016; 4.1 vs. 4.5 in 2017), and PK recorded a maximum LAI close to nF (2.8 vs. 2.9 in 2016; 2.5 vs. 2.3 in 2.17). SNPK and NPK showed significantly higher LAI than PK and nF post-flowering. Compared to NPK, SNPK recorded higher LAI in 2016 and lower in 2017. PK and nF had similar LAI post-silking (0.73 vs. 0.87 at the final measurement in 2016; 0.81 vs. 0.67 in 2017).

### 3.4. Leaf Photosynthesis

Photosynthetic rates (Pn) in SNPK and NPK were similar at silking stage and at approximately 30 days after silking (DAS) in 2016; however, they were 17.3% lower in SNPK compared to NPK at 50 DAS. In 2017, Pn was 16.6% lower in SNPK than in NPK at silking stage; however, Pn of SNPK was significantly higher compared to NPK at approximately 50 DAS (19.6 vs. 16.5 μmol m$^2$ s$^{-1}$). On average, Pn of SNPK and NPK was significantly higher than PK and nF at three measurements. Pn of PK was similar to nF post-flowering (Figure 4).

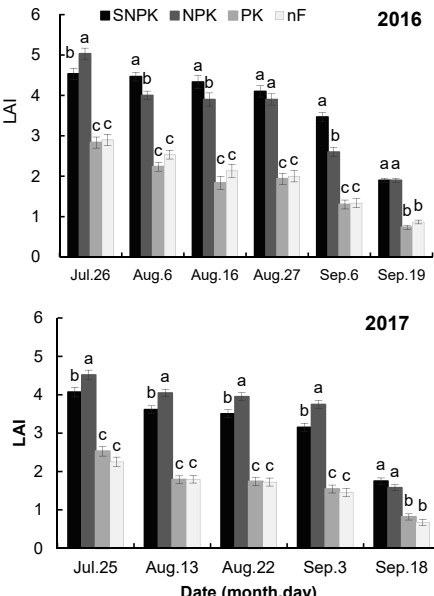

**Figure 3.** Leaf area index (LAI) post-flowering as effect of long-term fertilizer treatment SNPK, NPK, PK, and nF. SNPK represents combination of maize straw, chemical nitrogen, phosphorus, and potassium, NPK represents treatment of chemical nitrogen, phosphorus, and potassium, PK represents chemical phosphorus and potassium, and nF represents no fertilizer. Values with the same letter are not significantly different at $p < 0.05$, comparison within the same sampling date. Bars: Standard error of mean.

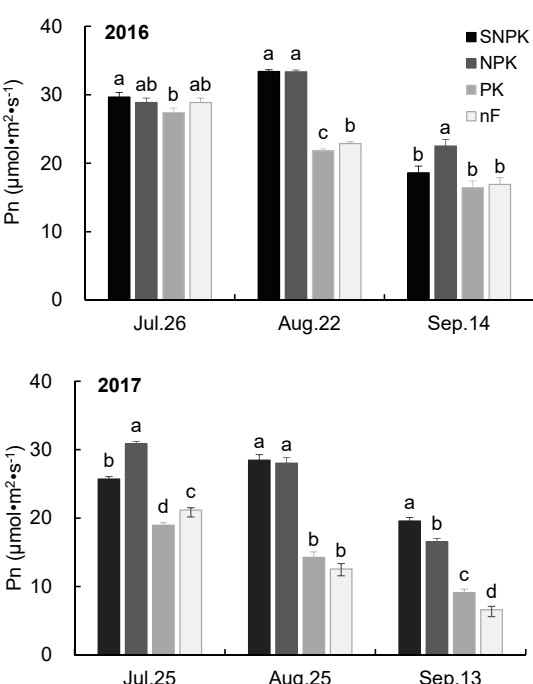

**Figure 4.** Photosynthesis rates (*Pn*) of maize ear leaf at the flowering stage (July 26 in 2016-A and July 25 in 2017-B), middle grain-filling stage (August 22 in 2016 and August 25 in 2017), and near maturity (September 14 in 2016 and September 13 in 2017) as an effect of long-term fertilizer treatment SNPK, NPK, PK, and nF. SNPK represents a combination of maize straw, chemical nitrogen, phosphorus, and potassium, NPK represents the treatment of chemical nitrogen, phosphorus, and potassium, PK represents chemical phosphorus and potassium, and nF represents no fertilizer. Values with the same letter are not significantly different at $p < 0.05$, comparison within the same measuring date. Bars: Standard error of the mean.

### 3.5. Post-Flowering Dynamics of Carbon, Nitrogen, Phosphorus, and Potassium in Leaf, Stem, and Grain

After flowering, total C concentration (%) decreased in leaf and increased in stem and grain, while N, P, and K concentrations decreased in leaf, stem, and grain (Figure 5). Total C concentrations (g plant$^{-1}$) remained stable in leaf and stem after flowering; however, N, P, and K concentrations decreased with time in both leaf and stem (Figure 6). In grain, C, N, P, and K concentrations increased during the entire grain-filling period (Figure 6A3,B3,C3,D3).

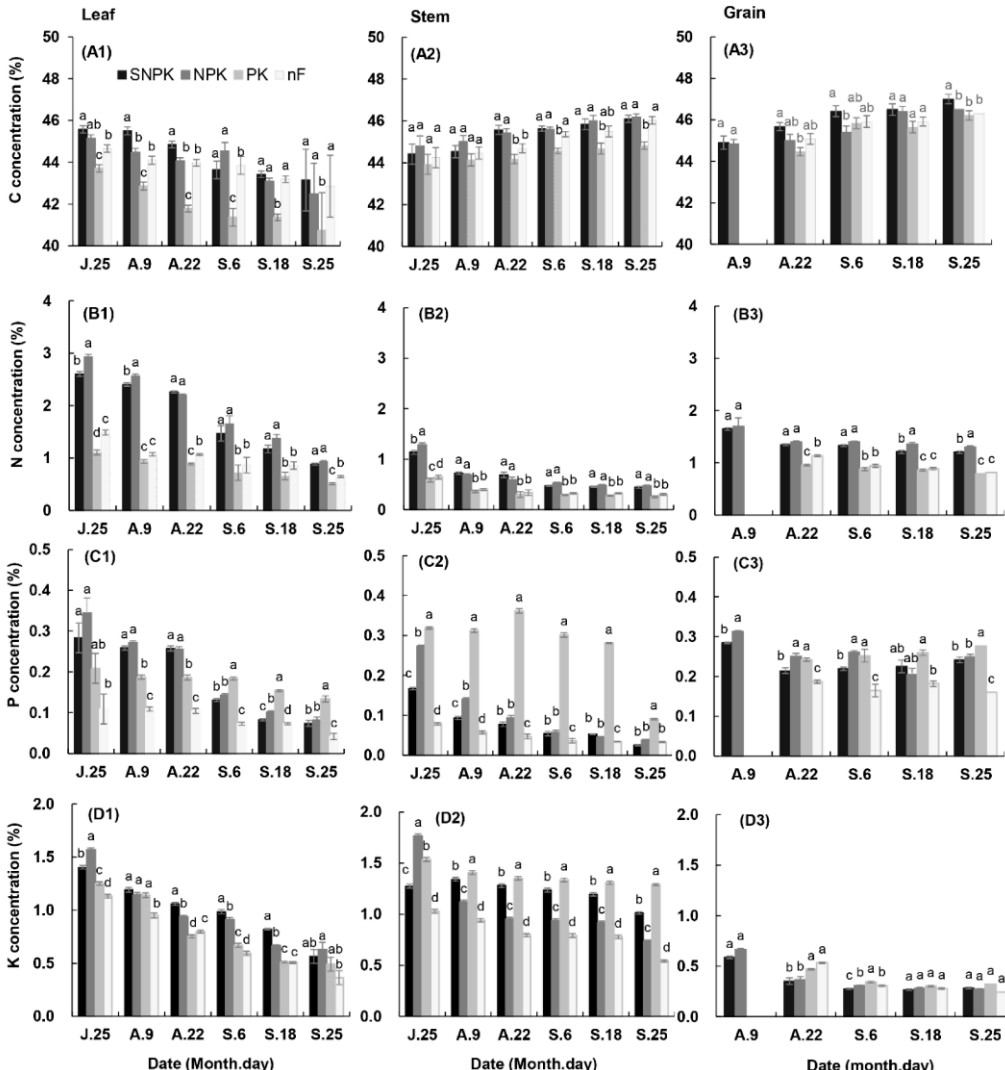

**Figure 5.** C, N, P, and K concentrations post-flowering in the leaf, stem, and grain in 2017 as the effect of long-term fertilizer treatment SNPK, NPK, PK, and nF. SNPK represents a combination of maize straw, chemical nitrogen, phosphorus, and potassium, NPK represents the treatment of chemical nitrogen, phosphorus, and potassium, PK represents chemical phosphorus and potassium, and nF represents no fertilizer. Values with the same letter are not significantly different at *p* < 0.05, comparison within the same measuring date. Bars: Standard error of the mean.

Total C, N, P, and K concentrations or contents in leaf, stem, and grain were similar in SNPK and NPK after flowering. In SNPK and NPK, these were higher than in PK and nF, especially for C and N content. PK had similar C and N content (Figure 6A1–A3,B1–B3) with the treatment of nF; however, had more P and K concentrations and contents in leaf and stem (Figure 5C1–C3,D1–D3; Figure 6C1–C3,D1–D3).

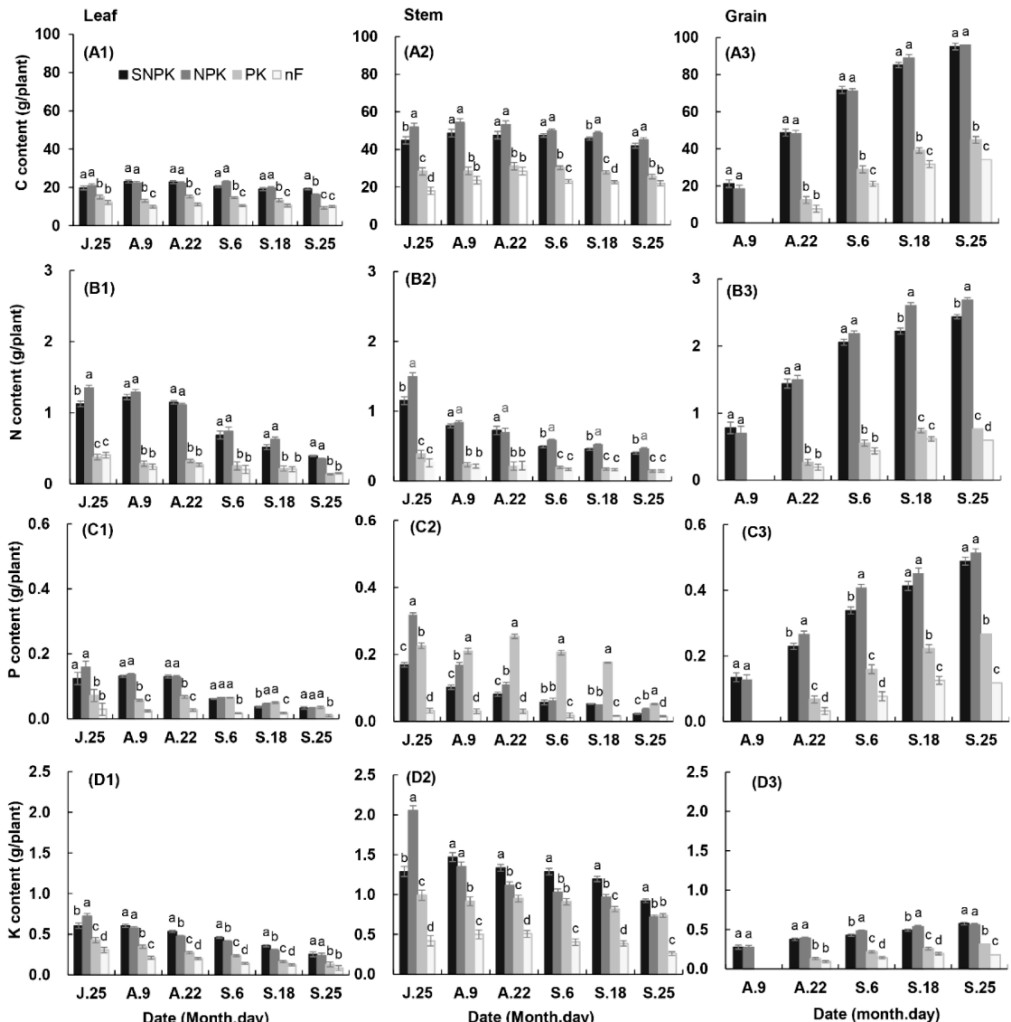

**Figure 6.** C, N, P, and K contents post-flowering in the leaf, stem, and grain in 2017 as the effect of long-term fertilizer treatment SNPK, NPK, PK, and nF. SNPK represents a combination of maize straw, chemical nitrogen, phosphorus, and potassium, NPK represents the treatment of chemical nitrogen, phosphorus, and potassium, PK represents chemical phosphorus and potassium, and nF represents no fertilizer. Values with the same letter are not significantly different at $p < 0.05$, comparison within the same measuring date. Bars: Standard error of the mean.

*3.6. Correlation between Post-Flowering Dynamics of Carbon, Nitrogen, Phosphorus, and Potassium and Grain Yield*

Grain yield, kernel number, and TKW significantly correlated with C, N, P, and K nutrients dynamics in leaf and stem after flowering (Tables 4 and 5). Nutrient dynamics included changes in concentration and content of C, N, P, and K at silking ($C_f$, $N_f$, $P_f$, and $K_f$), the respective mean value over the entire grain-filling period (C̄, N̄, P̄, and K̄), and the respective difference between silking and physiological maturity ($C_\triangle$, $N_\triangle$, $P_\triangle$, and $K_\triangle$).

For the dynamics of C, N, P, K concentration (%), grain yield, kernel number, and TKW significantly correlated with $N_f$, N̄, $N_\triangle$, $P_\triangle$, and K̄ in leaf (Table 4). TKW significantly correlated with Kf and $K_\triangle$ also in leaf. Grain yield, kernel number, and TKW significantly correlated with $N_f$, N̄, and $N_\triangle$, and C/N in stem.

For the dynamics of C, N, P, K content (gplant$^{-1}$), grain yield, kernel number, and TKW significantly correlated with Cf, C̄, $N_f$, N̄, $N_\triangle$, and C/N in both leaf and stem (Table 5). Grain yield, kernel number, and TKW significantly correlated with $P_\triangle$ and K̄ in leaf. In addition, both kernel number and TKW

significantly correlated with Pf and Kf, and TKW significantly correlated with $\overline{P}$ in leaf. Grain yield and yield components highly correlated with C, N, P, and K dynamics in leaf than in stem.

**Table 4.** The correlation analysis of C, N, P, K concentration (%) dynamics in leaf and stem post-flowering, grain yield, kernel number (kernel No.), and thousand kernel weight (TKW).

| | $C_f$ | $\overline{C}$ | $C_\Delta$ | $N_f$ | $\overline{N}$ | $N_\Delta$ | C/N | $P_f$ | $\overline{P}$ | $P_\Delta$ | $K_f$ | $\overline{K}$ | $K_\Delta$ |
|---|---|---|---|---|---|---|---|---|---|---|---|---|---|
| | | | | | | | Leaf | | | | | | |
| Grain yield | | | | * | * | * | * | | | * | | * | |
| Kernel No. | | | | * | * | * | | | | * | | ** | |
| TKW | | | | * | * | * | | | | ** | * | * | * |
| | | | | | | | Stem | | | | | | |
| Grain yield | | | | * | ** | * | * | | | | | | |
| Kernel No. | | | | * | * | * | * | | | | | | |
| TKW | | | | * | * | ** | * | | | | | | |

$C_f$, $N_f$, $P_f$, and $K_f$ represent C, N, P, and K concentrations at flowering stage, respectively. $\overline{C}$, $\overline{N}$, $\overline{P}$, and $\overline{K}$ represent mean C, N, P, and K concentrations over the grain-filling period. $C_\Delta$, $N_\Delta$, $P_\Delta$, and $K_\Delta$ represent C, N, P, and K changes in the period from flowering stage to maturity. C/N represents CN ratio. * and ** indicate significance at $p < 0.05$ and 0.01, respectively.

**Table 5.** The correlation analysis of C, N, P, K content (g plant$^{-1}$) dynamics in leaf and stem post-flowering, grain yield, kernel number (kernel No.), and thousand kernel weight (TKW).

| | $C_f$ | $\overline{C}$ | $C_\Delta$ | $N_f$ | $\overline{N}$ | $N_\Delta$ | C/N | $P_f$ | $\overline{P}$ | $P_\Delta$ | $K_f$ | $\overline{K}$ | $K_\Delta$ |
|---|---|---|---|---|---|---|---|---|---|---|---|---|---|
| | | | | | | | Leaf | | | | | | |
| Grain yield | * | * | | * | ** | * | * | | | * | | * | |
| Kernel No. | * | ** | | * | ** | * | * | * | | * | * | ** | |
| TKW | * | * | ** | ** | * | * | * | * | * | * | * | * | |
| | | | | | | | Stem | | | | | | |
| Grain yield | * | * | | * | ** | * | * | | | | | | |
| Kernel No. | * | ** | | * | * | * | * | | | | | | |
| TKW | * | ** | | ** | ** | ** | * | | | | | | |

$C_f$, $N_f$, $P_f$, and $K_f$ represent C, N, P, and K concentrations at flowering stage, respecitively. $\overline{C}$, $\overline{N}$, $\overline{P}$, and $\overline{K}$ represent mean C, N, P, and K concentrations over the grain-filling period. $C_\Delta$, $N_\Delta$, $P_\Delta$, and $K_\Delta$ represent C, N, P, and K changes in the period from flowering stage to maturity. C/N represents CN ratio. * and ** indicate significance at $p < 0.05$ and 0.01, respectively.

## 4. Discussion

### 4.1. Straw Return and Grain Yield

This study confirmed that the combined application of organic and inorganic fertilizers can maintain or enhance productivity in the long run in maize of Northeast China, consistent with the previous findings of other regions [24,28,29]. In the present study, inorganic N input was reduced by nearly one-third in SNPK as compared with the treatment of NPK; however, grain yield and yield components were similar in SNPK and NPK 26–27 years after straw return (Table 3); plant nutrient dynamics, particularly for C, N, and C/N ratio in both leaf and stem, worked together to maintain the high yield (Tables 4 and 5). These results indicate that straw return can completely offset the negative effects of reducing N input on maize yield, which in turn helps to reduce the use of chemical N in agriculture. According to FAO (Food and Agriculture Organization of the United Nations, 2016), there was a decline in global consumption of N fertilizers in agriculture since 2014. However, still more than $3 \times 10^7$ tons of N was used for agriculture in 2016, ranking the first in the world [30]. Grain yield in maize was not continuously recorded for 26–27 years in the present study. Therefore, the exact year from which straw return started to completely replace the reduced chemical N input was not determined.

In an earlier experiment [28], combined application of animal manure and mineral fertilizer contributed to increased grain yield in sweet maize compared to chemical fertilizer alone in the first and second years after manure application; however, combined straw return and mineral fertilization did not have such effects, probably because of the slow release of nutrients from straw [31,32]. In a long-term field experiment of 22 years, straw return alone resulted in grain yield similar to no fertilizer in the long run [11]. When straw was combined with chemical fertilizer, grain yield was as high as optimal chemical fertilization in the first year after straw return [11,33]. Supplementing/combining chemical fertilizer accelerates the decomposition and nutrient release from straw, which is dependent on soil physical and chemical properties [34]. Therefore, combined application of straw and inorganic fertilizer is recommended to manage crop residues.

### 4.2. Relationships between Leaf Area Dynamics and Plant Productivity, Plant Nutrient Dynamics, and Straw Return

SNPK treatment resulted in similar dry matter, leaf area dynamics, photosynthesis rate (Pn), and dynamics of N, P, and K in leaf, stem, and grain with NPK. SNPK produced higher dry matter compared to NPK during the entire growing period (Figure 2), which reflects persistent biomass production capacity of SNPK. Increase in crop productivity may be due to increased soil organic matter in SNPK [11,29]. However, the difference in LAI between SNPK and NPK was not the same in 2016 and 2017, probably due to the difference in rainfall distribution during two maize-growing seasons. Rainfall during May to July, the vegetative growth stage at which LAI was determined, was more in 2017 (Figure 1B). Wet soil enhances the decomposition and nutrient release from straw, thus improving leaf development and productivity [35,36]. This explains the increased leaf Pn at the late growth period (Figure 4) and TKW (Table 2) in SNPK compared to NPK in 2016. Therefore, accelerating straw decomposition at crop early vegetative growth stage will be an optimal strategy to manage crop straw in maize production.

Straw return increased C concentration in maize leaf and grain and also increased K concentration and content in stem (Figures 4 and 5), which may be due to an increase in soil organic C sequestration [22,37] and soil potassium levels [38]. Previous studies showed that straw return either alleviates soil N depletion [36] or increases soil N content [18]. In the present study, long-term straw return had no effects on P uptake in leaf, stem, and grain, even though straw return increased the potential P input. Wang et al. [39] and Hu et al. [40] have shown that straw retention promotes P leaching in soil. P content of straw used in the present study was low, i.e., 0.16%, which is not likely to offset the leached P in the slow process of decomposition. However, evidence is particularly limited. Detailed research is needed regarding P dynamics in the soil–plant system after straw return to evaluate the effects on enhancing soil quality and maintaining crop yield.

### 4.3. Long-Term Application of PK Fertilizers and No Fertilizer

Effects of long-term application of PK alone on leaf productivity and grain yield in maize were insignificant compared to no fertilizer (nF) in the present study. This reflects the importance of combined NPK fertilization in maintaining sustainable grain production. In the study by Zhang et al. [33], a 16-year experiment with the application of PK showed modest yield increase over time; however, the yield was similar to no fertilizer treatment and much lower than NPK and NP. In the present study, no straw was returned to the field in 26–27 years in PK and nF, which may have resulted in soil nutrient depletion through crop harvest and residue removal [41], soil erosion [42], and leaching [43]. Another study on long-term application of organic manure and N fertilizer showed a decrease in soil N content and soil organic matter over time [33,44], which caused yield decline [11]. Collectively, it indicates that combined fertilizers application and soil organic matter are the key factors that determine agricultural sustainability.

Based on the nutrient concentration, long-term application of PK fertilizers is not favorable for C and N uptake in maize. P and K were distributed in the stem in PK and remained stable throughout

the reproductive phase. In leaf, P remained stable while K decreased rapidly during grain filling. In all the treatments, K concentration and content decreased in both leaf and stem and increased slightly in grain. This indicates K loss from maize after flowering [45]. In the grain of PK, P and K concentrations were highest; however, their contents were low compared to SNPK and NPK. These results indicate that soil nutrients can be absorbed into the plant in the long-term application of PK fertilizer; however, most of nutrients are stored in stem rather than in grain because of the limited number of kernels.

*4.4. Relationship between Grain Yield and Plant Nutrient Dynamics*

The close correlation between N and grain yield indicates that N concentrations and contents and their dynamics in both leaf and stem after flowering play significant roles in determining grain yield relative to C, P, and K (Tables 4 and 5) in maize. Maintenance of high N content and N dynamics seem to be critical to increase maize yield, because N content determines plant photosynthetic capacity and N dynamics determines biomass remobilization [45,46]. Meanwhile, P contents and dynamics were more correlated with grain yield and yield components in leaf than in stem. These were more correlated with thousand kernel weights than with kernel number. These results reflect the importance of P in maintaining leaf productivity and grain yield in maize [47]. In addition, K content was more correlated with grain yield and yield components in leaf than in stem, suggesting the role of K in improving grain yield mainly by enhancing leaf productivity. Different from N and P, K is involved in the physiological processes related to abiotic stress (drought and cold) tolerance [48,49]. In the present study, the crop faced low temperature stress during the late grain-filling period, during which leaf K may have alleviated the effects of cold stress and improved yield. Furthermore, the significant correlation between yield and C content indicates that C in leaf and stem after flowering regulates yield (Table 5). The different patterns of nutrient dynamics (plant C, N, P, and K) post-flowering and their correlation with yield and yield components emphasize the significance of straw return in maintaining or increasing crop yield and reducing chemical fertilizer input.

## 5. Conclusions

Long-term combined fertilizer application can alter nutrient dynamics in different plant organs and biomass productivity, thus influencing grain yield and yield components in maize. Taking into consideration the total N input, a proper mixture of crop straw return (organic N) and N fertilizer can replace N fertilizer alone in the long run. The long-term PK fertilizer input (deficiency of soil organic matter and N input) did not have a significant effect on yield compared with no fertilizer input. Therefore, to reduce N, P, and K fertilizer input for maize production in China and beyond, we have to use the optimum/proper amount of fertilizers and maintain nutrient combined application through straw return in a long run.

**Author Contributions:** Designed experiment and prepared original draft: Y.L., L.W., and P.Z.; Conducted experiment: Y.L. and Y.W.; Collected and analyzed data: Y.L.; Review and editing: Y.L. and Y.W.

**Funding:** This research was funded by the National Key Research and Development Program of China (2017YFD0300603 and 2017YFD0300303), the National Natural Science Foundation of China (31701349), and the Agricultural Science and Technology Innovation Program of Jilin Province (Post-doctor project 188317).

**Acknowledgments:** We would like to thank the field managements for Hongjun Gao for experiments.

**Conflicts of Interest:** The authors declare no conflict of interest.

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
