# Peer review of "Straw Return with Reduced Nitrogen Fertilizer Maintained Maize High Yield in Northeast China"

_agronomy, doi:10.3390/agronomy9050229_

Round 1
Reviewer 1 Report
Please see attached pdf for my suggestions to improve the manuscript.

Author Response
1. Questions/Comments:
Is your experiment long term? I do not think so. You present only 2 years of data.
1. Response:
We thank reviewer for encouraging comments and critical suggestions for improvement of the manuscript. Reviewer’s comments are well considered. But it is true that our research was conducted based on the long-term field experiment over 25 years. In the current MS, we evaluated the effects of straw return and combined nitrogen, phosphorous and potassium application on the maize yield, leaf function, and nutrient accumulations in Northeast China under the long-term experiment over 25 years.
To avoid confusing, we changed the title into “Straw return with reduced NPK fertilizer improved maize yield in Northeast China”
2. Questions/Comments:
Please change title too <=12 words it’s too long. Also reduce abstract to 250 words.
2. Response:
Thanks for the suggestions. The title and abstract were made shorter; title has 12 words, and abstract has less than 250 words.
3. Questions/Comments:
Have you calculated fertilizer use efficiency?
3. Response:
We truly thank the reviewer pointed the error. We revised it as follows:
Page 1 Lines 20-22: These results indicate that combined nitrogen, phosphorus and potassium application is critical for increasing grain yield in maize. Combined fertilizers application and crop straw return will help improve yield and reduce the N fertilizer rate in the long run.
The sentence was rephrased.
4. Questions/Comments:
what about CN ration of straw resulting in higher demand of fertilizer N to break the straw residue. Please explain?
4. Response:
This is a very nice suggestion. Admittedly, N fertilizer application can accelerate straw residue decomposition, but it is a very complex process involving many factors such as temperature, soil moisture, soil texture, microorganism, etc, so we did not focus on CN ration of straw so closely. Instead, we focused on organic N content of straw to detect whether the organic N can replace the same amount of inorganic N in maintaining crop growth and productivity in the long run. This is also one of the most important objectives in our study. The relevant explanation was added.
5. Questions/Comments:
hybrid company name or provider name missing
5. Response:
We very thank the reviewer pointed the missing. And we added the hybrid provider name.
Page 2 Lines 30-31: Maize hybrid Zhengdan958 (provided by Institute of Crop Science, Chinese Academy of Agricultural Science) was used in the study, which has been a favorable maize hybrid for more than 10 years in China.
6. Questions/Comments:
what was the crop rotation?
6. Response:
We truly appreciate the reviewer’s comments. And we supplied it in the Materials and Methods part.
Page 3 Lines 5-6: We employed a randomized complete block design for the experiment under corn continuous cropping system with four fertilizer treatments as follows:
7. Questions/Comments:
Give P and K values not P2O5 and K2O
7. Response:
Reviewer’s question is properly considered. But in the researches on the plant nutrition management, P2O5 and K2O are determined to represent the available nutriments content. References:
[1] Gouzaye A., Epplin F. M., Wu Y., Makaju S. O. Yield and nutrient concentration response to switchgrass biomass harvest date. Agronomy Journal, 2014, 103 (6): 793-799.
[2] Adak T., Singh S., Sachan R. S. Influence of integrated management of bio-fertilizers and chemical fertilizers on post-harvest soil fertility status in fenugreek on a mollisol. Environment and Ecology, 2006, 24 (4): 796-802.
[3] Norwood III V. M., Kohler J. J. Organic reagents for removing heavy metals from a 10-34-0 (N-P2O5-K2O) grade fertilizer solution and wet-process phosphoric acid. Fertilizer Research, 1990, 26: 113–117.
8. Questions/Comments:
why not the P and K fertilizer rate adjusted? why only N? Please explain?
8. Response:
This is also a very nice suggestion. In our study, the amount of straw return was calculated based on N demand due to the importance of N fertilizer. P and K of straw were considered less because the low P content of straw and sufficient K content in soil in the experimental region (see Table 1 and the reference below). In fact, it is too difficult to consider NPK together when applying straw. Hopefully, this explanation can be accepted.
The explanation was added as the note of Table 2.
References:
[1] Wang X. F., Zhang K., Wang L. C, Zhang G. G., Xie J. G. Manage and control potash fertilizer scientifically and realize the high and stable yield for maize. Journal of Maize Science, 2004, 12(3): 92-95.
[2] Wang L. C, Wang Y. J., Zhu P., et al. Theory and practice of Jilin maize yield. China Science Press: Beijing, China, 2013.
9. Questions/Comments:
Change to Mg ha-1
9. Response:
We would like to accept the reviewer’s suggestion and revised it as 7.57 Mg ha-1.
10. Questions/Comments:
what was the size of each plot and how many replicates were there for your experiment. what was seeding rate? space issue
10. Response:
We would like to accept the reviewer’s suggestion and revised it.
Page 4 Lines 13-15: precision seeder, which maintained single grain sowing. Plots were south-north row orientation and the final stand of 60, 000 plants ha-1. Each plot consisted of 16 rows with 50 m long and 0.65 m inter-row spacing. Plants were sampled from the four center rows.
11. Questions/Comments:
why 14 percent? Usually standard is 15%
11. Response:
Reviewer’s question is properly considered. We would like to explain the 14% moisture content is the China national standard, and 14% moisture content is the US national standard.
12. Questions/Comments:
why this is figure 4? where are figure 2 and 3. Figures and table should appear in order. Also the second bar graph has missing legend on y axis. Check your standard error bars and be consistent with them
12. Response:
We would like to accept the reviewer’s suggestion and corrected the error.
13. Questions/Comments:
Remake these graphs as bar graphs and show letters on bars. If you want to use lines in graphs then please give LSD.
See my previous comment and change graph accordingly. It’s hard for the reader to make sense out of it. It’s too messy.
change figure 2 and 3 both to bar graphs
13. Response:
Thanks for the suggestions. The figures were changed into bar graph.
14. Questions/Comments:
It would be interesting to see CN ratio values and make comparison of those. I suggest you run your data for CN ratio and report that here.
14. Response:
We accepted the reviewer’s suggestion. Information about C/N were added in the text.
15. Questions/Comments:
I would like to see CN ration over here and how that play important role?
15. Response:
We accepted the reviewer’s suggestion.
Information about C/N was added in the results. And further discussion was added in discussion as follows:
Plant nutrient dynamics, particularly for C, N, and C/N ration in both leaf and stem, worked together to maintain the high yield (Table 4 and 5).
16. Questions/Comments:
FAOSTAT what is this?
16. Response:
We would like to accept the reviewer’s suggestion and revised it as FAO (2016). FAOSTAT is a database where crop yield can be downloaded. http://www.fao.org/faostat/zh/#compare
17. Questions/Comments:
what about P still present in undecomposed maize straw?
17. Response:
We accepted the reviewer’s suggestion and revised this sentence as:
Wang et al. [39] and Hu et al. [40] have shown that straw retention promotes P leaching in soil. P content of straw used in the present study was low, i.e. 0.16%, which is not likely to offset the leached P in the slow process of decomposition.
Evidences are very limited concerning straw P dynamics after straw return.
18. Questions/Comments:
I dont agree. Its seems you are over stretching your conclusion.
18. Response:
We truly appreciate the reviewer’s comments. We revised the expression and now it reads:
Long-term combined fertilizers application can alter nutrient dynamics in different plant organs and biomass productivity, thus influencing grain yield and yield components in maize. Taking into consideration the total N input, a proper mixture of crop straw return (organic N) and N fertilizer can replace N fertilizer alone in the long run. The long-term PK fertilizer input (deficiency of soil organic matter and N input) did not have a significant effect on yield compared with no fertilizer input. Therefore, to reduce N P and K fertilizer input for maize production in China and beyond, we have to use optimum/proper amount of fertilizers and maintain nutrient combined application through straw return in a long run.
19. Questions/Comments:
you did not measure air pollution
19. Response:
We would like to accept the reviewer’s suggestion and deleted this sentence in the conclusion.
Thank you again.
Reviewer 2 Report
There are alot of studies available in literature on the chemical fertilizer use with with manure. So, I dontthink this study is adding new information to literature. Also, authors needs to improve their manuscript, rearrange figures etc. Figure 2 and 3 in the manuscript are provided after figure 4, 5, 6 etc.
In results section, authors mostly write one treatments is significantly greater than other treatments, instead of providing a number for how much increase or decrease happened. Only writing "significantly higher or lower" is not providing much information to readers.

Author Response
1. Questions/Comments: What is meant by balanced here? 1. Response: We thank the reviewer 1 for the right comments and critical suggestions on the basic and important statement. We have revised it as ‘combination of NPK fertilizers’ and addressed all of your concerns suggested in the current revision. Here, we would like to show you the point-wise clarifications to the questions/comments that are presented as follows: Page 1 Lines 20-21: These results indicate that combination of NPK fertilizers is critical for increasing grain yield in maize. Page 1 Line 23: Combined fertilizer application; Page 2 Lines 13-14: growth and productivity, in the comparison between the scenarios of straw + reduced NPK fertilizers and NPK fertilizers. Page 2 Line 19: NPK and PK fertilizer treatments, combination of straw return and NPK fertilizers, and no fertilizer control. Page 3 Lines 7-8: potassium fertilizers added (Straw + NPK fertilizers); (2) nitrogen, phosphorus, and potassium fertilizers added (NPK); (3) phosphorus and potassium fertilizers added (PK); and (4) no fertilizer added (nF). Page 13 Line 10: Collectively, it indicates that combined fertilizers application and soil organic matter Page 13 Line 41: Long-term combined fertilizers application can alter nutrient dynamics in different plant Page 13 Line 48: combined application through straw return in a long run. 2. Questions/Comments: What is balance or unbalanced fertilizers? 2. Response: We thank reviewer 1 for encouraging comments and critical suggestions on the previous submission. As mentioned above, balanced fertilizer means a combination of NPK fertilizers. The terms “balance and unbalanced” are confusing, so we changed them into ‘combined fertilizers application’, and ‘unbalance fertilizers’ i.e. ‘PK fertilizers’. 3. Questions/Comments: I think it is better to write this treatment as straw + NPK fertilizer. When you write chemical NPK. I don’t get enough information what are saying. You need to write NPK fertilizer/s. Writing chemical NPK is not sufficient and correct. 3. Response: Reviewer’s comments are well considered. We have presented all SNPK as straw + NPK fertilizer and chemical NPK as NPK fertilizers in the current revision. 4. Questions/Comments: Which two years…you did not talk about these years in previous sentences. 4. Response: We thank for the reviewer’s constructive suggestions. Information about experimental years 2016 and 2017 were added in the abstract. 5. Questions/Comments: Already said this previous sentence. No need to repeat. 5. Response: We would like to accept the reviewer’s right suggestions. The sentence was deleted. 6. Questions/Comments: Vague sentence. Well…yes plant will accumulate all these nutrients, nothing new in here. You need to specify in the objective section what measurements you are taking. 6. Response: It is true that total carbon (C), nitrogen (N), and phosphorus (P) got accumulated in the whole plant during the grain filling period, however, K was lost from leaf, stem, and grain. The sentence was rephrased as In contrast with Total carbon (C), nitrogen (N), and phosphorus (P), K was lost from leaf, stem, and grain. 7. Questions/Comments: not look right? applying fertilizers increases biomass production and increases organic carbon in soil. Check this sentence 7. Response: Thanks for the detailed suggestion. A few references mentioned that long-term fertilization can result in decrease in the total soil organic matter especially in the simple fertilization treatment (N, P, or K). But some evidences showed the opposite results. There seems no consensus concerning this topic. Therefore, we combined this sentence with the previous one, and deleted the content about soil organic matter. 8. Questions/Comments: u mentioned about this in previous 2-3 lines also. Please delete it at one place. 8. Response: We would like to accept the reviewer’s suggestion and delete it in the abstract. 9. Questions/Comments: provide some more soil description. only thing that I learn from black soil is that is black in color. no the information. 9. Response: It is extremely important to use the standard soil classification. And we explain that the black soil is one kind of soil types according to the Chinese soil classification system. It is similar to the vertosols and includes Chernozem and Phaoezem. In US, it is called Mollisols. Therefore, we provide the basic information on the black soil. 10. Questions/Comments: 5.0 °C, are you sure...? 10. Response: We checked the annual temperature data from the Jilin Province Meteorological Bureau carefully. The average annual temperature is 5.0 °C truly. And we revised as the mean precipitation and temperature during maize growing season were 550 mm and 17.0°C, respectively. (Fig.1, the data determined by the Field Weather Station from 1 April to 1 October). 11. Questions/Comments: when were these soil samples collected and how...what methods are used for analysis? were soils nutrient status optimum for crop production? 11. Response: We thank the reviewer 1 for the critical comments on the detailed description of the methods. We added the related information in the present revision. Replacement has been made. Page 2 Lines 28-29: Each field experiment plot was collected 5 replicates by soil drill before sowing. The soil total N, P, and K were determined using standard methods [26]. Reference 26: Lu R. Analytical methods of soil agro-chemistry; China Agriculture Science and Technique Press: Beijing, China, 1999. 12. Questions/Comments: provide amount of straw. 12. Response: The relevant information was added in Table 2; straw (7571 kg ha−1, N content: 0.7%). 13. Questions/Comments: provide source of NPK fertilizers. for example. N was applied through urea or AN etc. 13. Response: We appreciated to accept the reviewer’s suggestion and provided the source of NPK fertilizers. Page 3 Lines 13-15: N fertilizer (Urea) was applied at two-time points; one third was applied at sowing and the rest at 6-leaf stage, in both SNPK and NPK treatments. At sowing, 82.5 kg P2O5 ha−1 (Superphosphate) and 82.5 kg K2O ha−1 (Potassium chloride) were applied 14. Questions/Comments: provide units inside the table above numbers please 14. Response: We added the units inside the table 2. 15. Questions/Comments: how much straw was applied kg/ha? 15. Response: We added the straw amount 7571 kg ha−1 inside the table 2. 16. Questions/Comments: The average fresh ear weight (G) was calculated, by which, 20 ears whose weight was equal to 20×G were selected. not clear 16. Response: Page 5 Line 4: The average fresh ear weight (G) was calculated, by which, 20 ears were selected with total weight approaching 20×G. 17. Questions/Comments: always provide results quantitatively, not qualitatively. Write how much higher instead of significant higher. 17. Response: We truly appreciate the reviewer’s comments and revised it. The paragraphs in results were rephrased. Quantitative and qualitative description was added in the results. 18. Questions/Comments: when the interaction is not significant, then data should be combined over years and then, present it in the tables. same is true for the HI. At present, since each year have its own statistical letters for treatments. it looks like there was an interaction effect. 18. Response: It is a very nice suggestion. The overall mean values in yield, yield components, and HI were added in Table 3. Thank you again.
Round 2
Reviewer 1 Report
Authors addressed all comments!